# SHARELoRA: LESS TUNING, MORE PERFORMANCE FOR LoRA FINE-TUNING OF LLMS

## ABSTRACT

Fine-tuning large language models (LLMs) is prohibitively expensive, prompting the development of various parameter-efficient fine-tuning (PEFT) methods. These methods primarily focus on fine-tuning small, additional modules known as *adapters*, which account for only a small fraction of the total LLM parameters. One such method, low-rank adaptation (LoRA), has shown notable parameter efficiency while maintaining performance comparable to full fine-tuning. However, classical LoRA may still involve tuning more parameters than necessary given the intrinsic rank of pre-trained weights, as highlighted by prior work (Aghajanyan et al., 2020). Recent variants of LoRA aim to enhance fine-tuning performance, but they overlook the layer-wise redundancies that can be leveraged for more efficient weight sharing. In this work, we introduce SHARELoRA, a novel approach that further enhances parameter efficiency during LLM fine-tuning by leveraging redundancies in pre-trained model weights to share LoRA modules, thereby significantly reducing the number of trainable parameters. Specifically, SHARELoRA automatically identifies redundancies in the pre-trained weights and determines which LoRA adapters can share parameters. This is achieved by measuring the similarity between representations to assess information redundancy and using a greedy algorithm to maximize parameter sharing. We conducted extensive evaluations on the LLMs of the LLaMA family across benchmark tasks. Notably, SHARELoRA achieves better parameter efficiency, with up to a 23% reduction in the number of fine-tuned parameters while delivering performance comparable to or better than existing PEFT methods.

## 1 INTRODUCTION

Large language models (LLMs), *e.g.,* GPT-4 and LLaMA2, are at the forefront of advances in the field of machine learning (ML). These large models are pre-trained on vast datasets (*e.g.,* images or text corpora) and are subsequently fine-tuned for specialized tasks, demonstrating proficiency in domains such as natural language, image processing, and fundamental scientific discoveries (Bommasani et al., 2021; Touvron et al., 2023a; Singhal et al., 2022; 2023). These models, often referred as "base model", are pre-trained solely to predict the next token to generate from their entire vocabulary space (Touvron et al., 2023a; Penedo et al., 2023; Team, 2023). To employ the base model for real applications, *e.g.,* building chatbots, then often need further fine-tuning (*e.g.,* on multi-turn human-human or human-chatbot conversations) to follow specific human instructions or align with human preferences (Leike et al., 2018; Ziegler et al., 2019; Chung et al., 2022).

Fine-tuning such large-scale LLMs, however, presents a computational challenge due to the massive number of parameters. For instance, GPU memory must be sufficiently large to handle the fine-tuning process while marinating a reasonably large batch size. Parameter-Efficient Fine-Tuning (PEFT) methods have been proposed to tackle this challenge by allowing fine-tuning of only a small subset of LLM parameters or incorporating small adapter modules on top of the pre-trained model, while leaving the majority of the base LLM parameters frozen (Houlsby et al., 2019; Hu et al., 2021; Zaken et al., 2021; Zhang et al., 2023). These methods democratize LLM fine-tuning by making it feasible on commodity hardware. One popular PEFT method, LoRA, has been shown to effectively reduce the GPU memory requirement during LLM fine-tuning (Hu et al., 2021). LoRA achieves parameter efficiency by adding low-rank adapters in parallel with specific LLM parameters, such as the query,

| FT Method | GSM8K ↑ | ARC-Challenge ↑ | WinoGrande ↑ | Hellaswag ↑ |
|---|---|---|---|---|
| LoRA($r = 12$) | **37.98** | **48.21** | **64.25** | **51.96** |
| Naive-shared LoRA | 37.23 | 47.95 | 62.83 | 48.38 |

Table 1: Accuracy comparison of LoRA and a naive-share LoRA strategy.

key, and value parameter weights in multi-head attention. During fine-tuning, LoRA LoRA optimizes only the low-rank adapters, while the LLM parameters remain unchanged.

Although LoRA is efficient, it treats all layers uniformly, lacking finer control over which layers are most important or exhibit similar behavior. Recent improvements, such as AdaLoRA (Zhang et al., 2023), which dynamically adjusts the rank based on layer importance, DoRA (Liu et al., 2024a), which decouples weights into direction and magnitude for more nuanced fine-tuning, and LoRA+ (Hayou et al., 2024), which independently adjusts the learning rates of LoRA components, have aimed to enhance LoRA's efficiency. However, these approaches overlook the redundancy present in pre-trained foundation models, where certain layers may exhibit similar behavior and can potentially share parameters, further reducing memory requirements.

We observed that sharing the LoRA module's weights across layers does not significantly degrade performance, while effectively reducing the number of trainable parameters. Specifically, we experimented by sharing the LoRA weights between odd-numbered and even-numbered layers, which halved the number of trainable parameters. In this naive approach, the weights of even-numbered layers were directly mirrored to their adjacent odd-numbered layers. Surprisingly, this straightforward weight-sharing strategy led to only minor performance degradation, as demonstrated in Table 1. This finding suggests that there is significant potential for further optimization through weight sharing in LoRA modules. However, determining which layers should share weights remains an ongoing challenge, largely due to the limited explainability of foundation models (Zhao et al., 2024). The behavior and interaction of different layers within these models are not yet fully understood, and there is no consensus on which layers exhibit sufficiently similar representations to justify weight sharing. This is an active area of research, and more sophisticated methods for identifying redundant layers could lead to even more efficient weight-sharing strategies in the future.

In foundation models, certain layer representations often exhibit notable similarities. This redundancy is a result of insufficient training data, which prevents each parameter from learning distinct, unique features. Consequently, this leads to overlapping or redundant representations across layers. Prior research has harnessed these redundancies for model compression (Gromov et al., 2024). Building on this insight, our work leverages this redundancy by sharing LoRA weights across layers with similar representations during fine-tuning. This approach allows us to slightly increase the LoRA rank without increasing the overall number of trainable parameters, thereby enhancing fine-tuning performance.

Our SHARELORA method consists of two main components: (i) computing similarity matrices between representations of layers and (ii) sharing the LoRA module parameters among redundancy layers. This method identifies layers with similar representations and shares their weights, reducing the number of trainable parameters while maintaining model performance. By leveraging layer similarities, SHARELORA significantly improves finetuning efficiency. We conduct extensive experiments on a wide range of tasks and models to demonstrate the effectiveness of SHARELORA. Specifically, we evaluate the performance using LLaMA-7B, LLaMA2-7B, and LLaMA3-8B for natural language commonsense reasoning and LLava-1.5-7B for image-text understanding. The results show that SHARELORA performs similar or better than LoRA, while only using 80% trainable parameter.

The summary of our contributions is as follows:

- We propose SHARELORA, a novel parameter-efficient fine-tuning (PEFT) method that leverages the similarity of layer representations to enable weight sharing, achieving this without introducing any additional train or inference latency compared to LoRA.

- We develop a greedy algorithm to determine which layers should share the weights of the LoRA module based on the similarity of their representations.

- We conduct extensive experiments demonstrating that SHARELoRA consistently performs similar or better LoRA across various tasks, while using less trainable parameters.

## 2 PRELIMINARY AND MOTIVATION

**Parameter Efficient Fine Tuning.** A primary research trajectory aimed at reducing the fine-tune parameters of pretrained FMs is to model the incremental updates of pretrained weights in a parameter-efficient manner. For example, given a pretrained weight matrix $W$, diff pruning (Guo et al., 2021) initializes $\Delta$ as the same dimensions as $W$ and performs magnitude-based pruning on $\Delta$. Diff pruning characterizes $\Delta$ as the incremental updates of $W$, and it can improve the parameter efficiency due to the sparsity of $\Delta$. However, it requires specific hardware support to accelerate the computation of unstructured sparse matrices. This hardware-specific dependency underscores a crucial consideration for the practical deployment of such an approach in real-world applications. In addition, it does not significantly reduce computational cost compared to full fine-tuning (Hu et al., 2021), as every entry of $\Delta$ needs to be updated and then be pruned.

To tackle those limitations, Hu et al. propose LoRA (Hu et al., 2021), which parameterize $\Delta$ as the product of two low-rank matrices:

$$W' = W + \Delta = W + BA, \tag{1}$$

where $\Delta \in \mathbb{R}^{d_1 \times d_2}$, $B \in \mathbb{R}^{d_1 \times r}$, and $A \in \mathbb{R}^{r \times d_2}$ with $r \ll \{d_1, d_2\}$. As the rank $r$ is much smaller than the dimension of $W$ (e.g., $r = 8$ and $d_1 = d_2 = 4096$), the number of trainable parameters and training overhead dramatically decreases.

However, LoRA has its limitations, as it typically applies separate parameters for each $B$ and $A$ by default. This approach overlooks the significant variation in the redundancy of weight matrices across different layers and modules during the fine-tuning of pretrained foundation models. We will demonstrate this issue later. Consequently, LoRA cannot adaptively share the LoRA modules among the redundancy layers, which could otherwise achieve comparable performance with fewer trainable parameters.

**Weight Sharing.** Although PEFT methods can reduce the number of trainable parameters, thereby decreasing GPU memory footprint during fine-tuning, the number of parameters in LLMs also scales rapidly, making it increasingly challenging to conduct even PEFT on commodity GPUs. In this paper, we explore the possibility of combining LoRA with *weight sharing*, a method that allows multiple neural network layers to share the same model weights. Weight sharing has been a widely adopted technique to reduce the number of trainable parameters while maintaining performance and sometimes helping mitigate overfitting (Press & Wolf, 2017; de Lhoneux et al., 2018; Lan et al., 2020; Dai et al., 2020; Takase & Kiyono, 2021).

**Similarity Across Layers.** Our motivation for weight sharing among LoRAs across layers comes from the observation that the representations among model layers/blocks attain high levels of similarity, especially for bottom layer blocks (as shown in Figure 1). This effect has also been observed by many prior works (Kornblith et al., 2019; Gromov et al., 2024). This effect is also connected to methods like *layer skipping* and *mixture of depth* for LLMs (Fan et al., 2024; Elhoushi et al., 2024; Raposo et al., 2024). Previous research (He et al., 2024) demonstrates that the distribution of similarity scores remains consistent, even when the calibration dataset sample size varies. Additionally, prior studies indicate that altering the type of calibration dataset—whether it be pretraining datasets (e.g., C4 (Raffel et al., 2020)) or instruction-tuning datasets (*e.g.,*, CodeAlpaca-20k, MathInstruct (Xiang Yue, 2023), and LIMA (Zhou et al., 2024))—has minimal effect on the similarity score distribution.

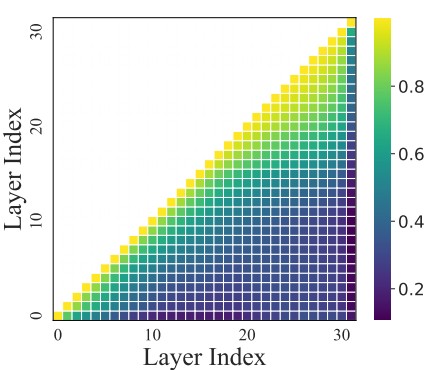

Figure 1: Cosine similarity among layers' representations experimented on LLaMA2-7B model over the GSM8k evaluation dataset.

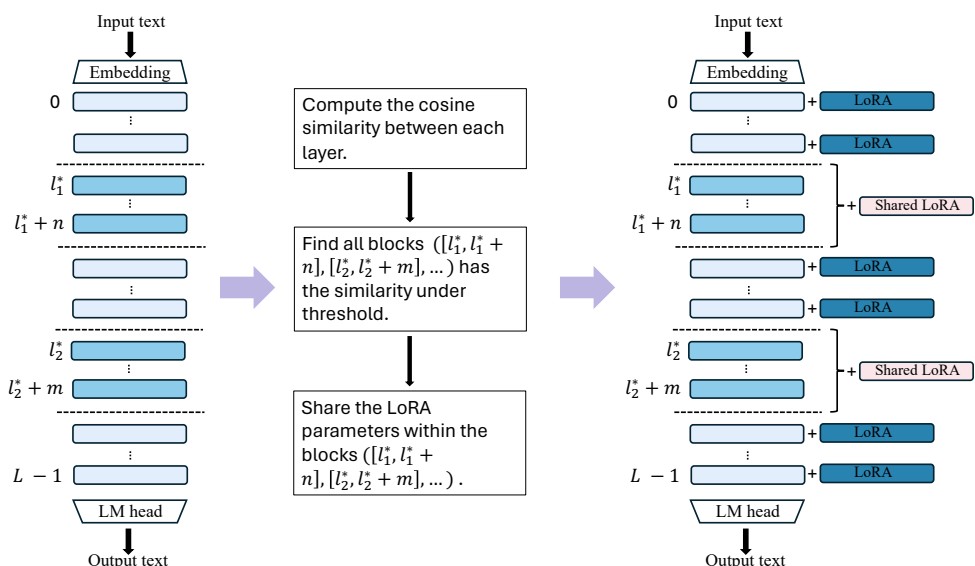

Figure 2: An overview of our proposed AutoLoRA, which computes the cosine similarity between each layer, and shares the LoRA parameters within redundancy blocks.

## 3 SHARELORA METHOD

As illustrated in Figure 2, our method comprises two key components: (i) layer similarity computation and (ii) LoRA sharing.

### 3.1 THE FORMULATION OF SHARELORA

In this subsection, we present the formulation of SHARELORA as an optimization problem. Specifically, we consider an $L$-layer LLM. We denote $\mathcal{S}$ as the collection of shared sets of LLM layer indices, *e.g.*, $\mathcal{S} = \{(2,3),(5,6,7),(11,13)\}$.

Our objective is to maximize the number of shared LLM layers while ensuring that the similarity measure between any two layers with shared weights exceeds a predefined threshold, $\epsilon$. This can be formulated as the following optimization problem:

$$\max \quad \sum_{\forall s \in \mathcal{S}} |s| \tag{2}$$

$$\text{s.t.} \quad c(\mathbf{x}_i, \mathbf{x}_i) \geq \epsilon \tag{3}$$

where $|s|$ is the number of layers in set $s$, and $\mathcal{S}$ is the collection of sets such that each set $s$ contains layers with similarity measure (*i.e.*, $c(\cdot, \cdot)$) between each pair exceeding the threshold $\epsilon$.

The objective of SHARELORA is to maximize the total number of layers sharing LoRA. The remaining question is *How to find $\mathcal{S}$?* We will present our algorithm of SHARELORA in section 3.3.

### 3.2 SIMILARITY BETWEEN REPRESENTATIONS OF LAYERS

To decide which layer's LoRA module could be shared with another, we have to compute the angular distance $c(\mathbf{x}_i, \mathbf{x}_j)$ between the representations of layer $i$ and layer $j$. The similarity of a single sequence of length $T$ is given by

$$c(\mathbf{x}_i, \mathbf{x}_j) := \frac{1}{\pi} \cos^{-1}\left(\frac{\mathbf{x}_i^\top \mathbf{x}_j}{\|\mathbf{x}_i\|\|\mathbf{x}_j\|}\right), \tag{4}$$

where the inner product is over the hidden dimension of the model for the final token $T$ of the sequence, $\|\cdot\|$ denotes the $\ell_2$-norm, and the factor of $1/\pi$ is a convention. This distance should then be summed over a number of examples that is large enough to get a low-fluctuation estimate but overall should be quite small.

## 3.3 SIMILARITY-BASED WEIGHT SHARING

The objective of the algorithm is to identify sets of neural network layers that can share LoRA modules based on the angular distance of their representations. The algorithm employs a greedy strategy to maximize the size of these shared sets while adhering to a predefined similarity threshold.

First, the algorithm processes the similarity matrix to create an upper triangular matrix, excluding the diagonal, which represents the pairwise similarities between layers. It then identifies all eligible pairs of layers $(i, j)$ where the similarity score exceeds the threshold $\epsilon$. Formally, this can be expressed as: shared_pairs $= \{(i, j) \mid c(\mathbf{x}_i, \mathbf{x}_j) \geq \epsilon \text{ and } 0 \leq i < j < L\}$.

Next, the algorithm constructs sets of shared layers by iterating through the identified shared pairs. It maintains a set visited to avoid reprocessing layers. The construction of the shared sets proceeds is shown in Algorithm 1.

---

**Algorithm 1** Construct Shared Sets

1: **Input:** Similarity matrix $C$, threshold $\epsilon$;
2: **Output:** Collection of shared sets $\mathcal{S}$;
3: Initialize shared_pairs $\leftarrow \{(i, j) \mid c(\mathbf{x}_i, \mathbf{x}_j) \geq \epsilon \text{ and } 0 \leq i < j < L\}$;
4: Sort shared_pairs by similarity scores;
5: Initialize $\mathcal{S} \leftarrow \{\}$, and initialize visited $\leftarrow$ set();
6: **for all** $(i, j)$ in shared_pairs **do**
7:     **if** $i$ not in visited and $j$ not in visited **then**
8:         **if** $i$ not in $\mathcal{S}$ **then**
9:             Initialize $S_i \leftarrow \{i\}$
10:         **end if**
11:         Add $j$ to $S_i$, and add $j$ to visited
12:         **if** All pairs starting with $i$ is done **then**
13:             Add $i$ to visited
14:         **end if**
15:     **end if**
16: **end for**

---

The algorithm ensures that layers are grouped into shared sets only if their pairwise similarities exceed the threshold $\epsilon$, thus maximizing the number of layers that can share the LoRA modules while maintaining high similarity within each set.

---

**Algorithm 2** SHARELORA

1: **Input:** Sample Dataset $\mathcal{D}^*$; Train Dataset $\mathcal{D}$; hyper-parameter similarity threshold $\epsilon$.
2: Inference on $\mathcal{D}^*$ and save the representations $x(l)$ for each layer $l$;
3: Compute the angular distance $d(x(l_p), x(l_q))$ between each layer $l_p$ and $l_q$;
4: Compute the share set $S$ and update LoRA rank $r$;
5: Share parameters using $S$;
6: Finetune weight sharing model $W$ with $\mathcal{D}$.
7: **Output:** The fine-tuned parameters $W^*$.

---

Furthermore, after sharing layers using the set $\mathcal{S}$, the number of layers with shared parameters will be $\sum_{S_i \in \mathcal{S}} |S_i| - |\mathcal{S}|$. This allows us to optionally expand the original rank $r$ to $\lfloor \frac{L}{L - \sum_{S_i \in \mathcal{S}} |S_i| + |\mathcal{S}|} r \rfloor$, where $L$ is the total number of layers. We summarize the algorithm in Algorithm 2.

| Model | PEFT Method | # Params | BoolQ | PIQA | SIQA | HS | WG | ARC-e | ARC-c | OBQA | Avg. |
|-------|-------------|----------|-------|------|------|-----|-----|-------|-------|------|------|
| | Prefix | 10.5M | 57.46 | 50.49 | 33.62 | 28.38 | 64.8 | 29.84 | 24.49 | 26.6 | 39.46 |
| | AdapterH | 20.1M | 71.19 | 74.48 | 45.45 | 57.24 | 59.51 | 56.90 | 33.70 | 39.0 | 54.68 |
| LLaMA-7B | AdapaterP | 20.1M | 67.65 | 73.45 | 44.27 | 57.04 | 58.48 | 57.62 | 33.87 | 37.0 | 53.67 |
| | Parallel | 448M | 73.36 | 74.54 | **73.81** | 57.08 | 60.22 | 56.23 | 35.58 | 36.8 | 54.70 |
| | LoRA($r = 8$) | 14.0M | **78.53** | 78.45 | 46.98 | **73.41** | 70.17 | **70.24** | **41.21** | 42.2 | 62.65 |
| | SHARELORA* | 12.1M | 73.94 | 79.38 | 47.08 | 72.93 | 68.11 | 65.99 | 37.8 | 43.0 | 61.03 |
| | SHARELORA(ours) | **10.8M** | 77.55 | **79.65** | 47.54 | 73.04 | **70.96** | 69.95 | 40.70 | **45.4** | **63.10** |
| LLaMA2-7B | LoRA($r = 8$) | 14.0M | 80.64 | 79.16 | **47.85** | **75.18** | 69.93 | **69.70** | **42.06** | 44.0 | **63.56** |
| | SHARELORA(ours) | **11.8 M** | **80.89** | **79.22** | 46.78 | 74.04 | **71.27** | 68.73 | 41.81 | 44.0 | 63.36 |
| LLaMA3-8B | LoRA($r = 8$) | 14.2M | **82.17** | 81.34 | **49.49** | 78.38 | **74.19** | 76.39 | **50.77** | **46.4** | **67.39** |
| | SHARELORA(ours) | **11.4M** | **82.17** | **81.56** | 48.77 | **79.32** | 73.80 | **76.85** | 50.00 | 44.2 | 67.07 |

Table 2: Accuracy with LLaMA model family with various PEFT methods on commonsense reasoning tasks.

# 4 EXPERIMENTS

**Models and datasets.** We implemented SHARELORA to fine-tune LLaMA family LLMs, namely, LLaMA-7B (Touvron et al., 2023a), LLaMA2-7B (Touvron et al., 2023b) and LLaMA3-8B (Meta, 2024). We follow the settings of LLM-Adapters (Hu et al., 2023), and evaluate the effectiveness of the several natural language commonsense reasoning task including BoolQ (Clark et al., 2019), PIQA (Bisk et al., 2019), SIQA (Sap et al., 2019), HellaSwag (Zellers et al., 2019), Winogrande (Sakaguchi et al., 2021), ARC-Easy (Clark et al., 2018), ARC-Challenge, and OBQA (Mihaylov et al., 2018). Additionally, we implemented SHARELORA to fine-tune LLaVA-1.5-7B (Liu et al., 2023), a popular vision language foundation model (VLM) and used on image-text pair understanding, and evaluated on LLaVA-Bench (in-the-wild) evaluation dataset (Liu et al., 2023).

**Setup.** We use PyTorch (Paszke et al., 2019) to implement all the algorithms. Our fine-tuning algorithm implementation is based on the publicly available Huggingface Transformers (Wolf et al., 2019) and LLM-Adapters code base. All the experiments are conducted on NVIDIA A6000 GPUs.

**Baselines.** We compare SHARELORA with the following baselines.

- *Prompt learning (Prefix):* (Li & Liang, 2021) Involves fine-tuning a small set of continuous task-specific vectors (prefixes) while keeping the large language model parameters frozen to pre-trained weights.
- *Adapter tuning (AdapterH):* (Houlsby et al., 2019) Inserts small add-on layers between the multi-head attention modules and FFN modules of the pre-trained model that can be fine-tuned for downstream task learning, while keeping the rest of the model parameters frozen.
- *Pfeiffer adapter (AdapterP):* (Pfeiffer et al., 2020) Unlike adapter tuning it inserts add-on layers after FFN modules and LayerNorm modules, allows them to fine-tune and keeps the rest of the model frozen.
- *Parallel adapter (Parallel):* (He et al., 2021) Modifies the hidden representations in a transformer model by inserting additional trainable parameters in parallel to the original model's layers.
- *LoRA:* (Hu et al., 2021) Is the most popular PEFT method that injects trainable low-rank matrices into transformer layers parallel to the frozen main path, to approximate the weight updates.
- SHARELORA* is a variation of our SHARELORA method, but instead of using similarity scores to select the set $S$, it randomly selects adjacent layers for weight sharing.

## 4.1 EVALUATIONS ON LLMS

We assess the fine-tuning performance of LLaMA-7B, LLaMA2-7B, and LLaMA3-8B using all baseline methods along with the proposed algorithm. The commonsense reasoning evaluation includes eight sub-tasks, each with its own predefined training and testing sets. Following the setup from LLM-Adapters (Hu et al., 2023), we combine the training datasets from all eight tasks to form a comprehensive training dataset for fine-tuning and performing evaluations on the individual testing sets for each sub-task.

**Implementation details.** All of LLaMA models have 32 hidden layers. Initially, we calculate the similarity $c(\mathbf{x}_i, \mathbf{x}_i)$ between each layer's representation using 256 randomly sampled C4 validation dataset (Raffel et al., 2020). We set the similarity threshold $\epsilon$ to 0.85 for LLaMA-7B and LLaMA2-7B, and 0.80 for LLaMA3-8B. Utilizing our similarity-based weight-sharing algorithm, the share set collection $\mathcal{S}$ for LLaMA-7B is $\{\{16, 17\}, \{18, 19\}, \{20, 21, 22\}, \{23, 24, 25, 26, 27\}, \{28, 29, 30\}\}$, enabling the sharing the LoRA modules of **ten** similar layers. The share set collection $\mathcal{S}$ for LLaMA2-7B and LLaMA3-8B are $\{\{17, 18\}, \{19, 20\}, \{21, 22, 23\}, \{24, 25, 26, 27\}, \{28, 29\}\}$ and $\{\{19, 20\}, \{21, 22\}, \{23, 24, 25\}, \{26, 27, 28\}\}$ separate. Note that we expand the original LoRA rank from 8 to 9 for these three models. The hyperparameters for fine-tuning across LLaMA-7B, LLaMA2-7B, and LLaMA3-8B models are consistent for most settings. A dropout rate of 0.05, AdamW optimizer, a linear learning rate scheduler, a batch size of 16, and 3 epochs are applied for all models. The learning rate (LR) is set to 2e-4 for LLaMA-7B and LLaMA2-7B, while it is reduced to 1e-4 for LLaMA3-8B. The fine-tuning targets the Q, K, V, Down, and Up layers for each model. The share set collection $\mathcal{S}$ for SHARELORA* is randomly generated as $\{\{2, 3\}, \{4, 5, 6\}, \{9, 10\}, \{11, 12, 13\}, \{14, 15\}, \{19, 20\}, \{22, 23\}, \{29, 30\}\}$.

Table 2 shows experimental results on the eight commonsense reasoning tasks. SHARELORA achieves similar or better performance across all datasets for all the models. Notably, our proposed approach saves up to **23**% of the trainable parameters yet achieves a **1.5**% improvement in performance with LLaMA-7B, while maintaining similar performance on LLaMA2-7B and LLaMA3-8B.

## 4.2 EVALUATIONS ON MULTI-MODAL VLMS

We now present the results of SHARELORA on vision-language models. We used the LLaVA-1.5-7B (Liu et al., 2023), which consists of a language model, a visual encoder and a projection layer for feature alignment. Specifically, the language model and visual encoder were initialized with Vicuna-1.5-7B (Zheng et al., 2024) and CLIP ViT-L/336px (Radford et al., 2021), respectively. In addition, we employed the pretrained projection layer[1] and directly performed visual instruction fine-tuning stage. In contrast to the setup of LLaVA, we chose a subset of their dataset, i. e. LLaVA-Instruct-80K. It consists of 80k image-instruction pairs filtered out from LLaVA-Instruct-150K[2]. We set the rank to

|  | LoRA(rank=64) | SHARELORA |
|---|---|---|
| # Params | 159.9M | **139.4**M |
| Description | $45.57 \pm 0.60$ | **45.80** $\pm 0.40$ |
| Conversation | $46.87 \pm 1.27$ | **54.10** $\pm 0.00$ |
| Reasoning | **66.90** $\pm 0.85$ | $65.37 \pm 0.72$ |
| All | $55.47 \pm 0.55$ | **57.00** $\pm 0.44$ |

Table 3: Instruction-following capability comparison between LoRA and SHARELORA (ours). We conduct three repeated evaluations and report the average scores. The results are reported in the format of $mean \pm std$.

64 and performed instruction tuning for one epoch of the LLM component using SHARELORA and LoRA, respectively. Note, we keep the feature transformation module trainable for both.

**Implementation details.** For SHARELORA we compute the similarity between the inputs and outputs of each layer based on 128 samples, with each sample comprising of visual-text inputs. The similarity threshold was set to 0.9 and the share set collection $S$ for language model is $\{\{2, 3\}, \{4, 5\}, \{6, 7\}, \{17, 18\}, \{19, 20\}, \{21, 22, 23\}, \{24, 25, 26\}, \{27, 28, 29\}\}$. The hyperparameters used for fine-tuning both LoRA and SHARELORA on the vision-language model are as follows. For the similarity threshold, only SHARELORA utilizes a value of 0.9, while LoRA does not incorporate this parameter. Both methods share the same rank $r = 64$, scaling factor $\alpha = 128$, and a dropout rate of 0.05. The target layers include Q, K, V, O, Up, Down, and Gate for both methods. The training runs for one epoch with a learning rate of 2e-4, and the learning rate is adjusted using a cosine scheduler. Both approaches employ the AdamW optimizer, with a batch size of 64 and a warmup ratio of 0.03. These shared hyperparameters ensure consistent conditions for comparing the performance of LoRA and SHARELORA. We use GPT-4 generated answers as the golden answers. Subsequently, we employed GPT-4o to evaluate the instruction-following capabilities of the fine-tuned model and selected the challenging LLaVA-Bench (in-the-wild).

---

[1]https://huggingface.co/liuhaotian/llava-v1.5-mlp2x-336px-pretrain-vicuna-7b-v1.5
[2]https://huggingface.co/datasets/liuhaotian/LLaVA-Instruct-150K

It comprises 24 images with a total of 60 questions. In Table 3, we demonstrate the performance of SHARELoRA and LoRA fine-tuned LLaVA model. Overall, SHARELoRA exhibits superior performance compared to that with LoRA, with a significant advantage in the conversation aspect. Additionally, SHARELoRArequires around **13**% fewer trainable parameters to yield this improved performance.

### 4.3 ABLATIONS AND DISCUSSIONS

**Robustness towards different rank settings.**

Here we investigate the impact of various rank configurations on SHARELoRA and LoRA by adjusting the original $r$ within the set $\{2, 4, 8, 16\}$. We then evaluate the performance of fine-tuned LLaMA-7B on commonsense reasoning tasks as described in §4.1. Fig. 3 depicts the results with different ranks for both LoRA and SHARELoRA. Across all four original rank settings, SHARELoRA consistently improves performance compared to the baseline LoRA. Despite using the same original rank, SHARELoRA employs only 80% of the trainable parameters relative to the LoRA. For instance, SHARELoRA achieves an average accuracy of 63.35% on the eight commonsense reasoning tasks with 5M trainable parameters, whereas LoRA requires 7M parameters for the same rank level.

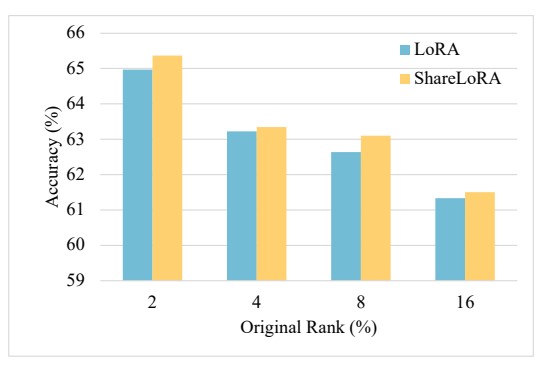

Figure 3: Accuracy comparison with different rank values for LoRA and SHARELoRA.

**Similarity threshold.**

Table 4 presents the experimental results of fine-tuning LLaMA-7B with different similarity thresholds $\{0.75, 0.80, 0.85, 0.90\}$. The best performance is achieved when the similarity threshold $\epsilon$ is set to 0.85. Higher similarity thresholds result in more eligible shared layers. As the number of shared layers increases, the rank of the LoRA modules can be adjusted upward within the constraints of the trainable parameter budget, which is determined by the original rank. Thus, there is a tradeoff between having more independent layers and a larger rank. Consequently, selecting an optimal similarity threshold $\epsilon$ is crucial for balancing the trade-off between the number of shared layers and the rank of LoRA modules. Higher thresholds allow more layers to share weights, permitting an increase in the rank within the limits of the trainable parameter budget. Therefore, the choice of $\epsilon$ significantly influences the method's performance.

| Similarity threshold | Accuracy |
|---|---|
| 0.75 | 61.80 |
| 0.80 | 62.81 |
| 0.85 | 63.10 |
| 0.90 | 61.80 |

Table 4: Accuracy comparison of different similarity thresholds evaluated with LLaMA-7B.

In practice, we typically choose $\epsilon$ between 0.80 and 0.90, resulting in about one-third of the layers sharing weights with others. We can also run Algorithm 1 independently to obtain the shared dictionary, helping select an appropriate $\epsilon$ before fine-tuning.

**Adapter sensitivity to different layer types.**

In subsection 4.1, we adapt our proposed method to the Q, K, V, Down, and Up weights. Table 5 presents the performance of fine-tuning the weights associated with each module type separately. We use $r = 8$ for the LoRA modules. The results indicate that adapting our proposed method to the Q and K weights yields the most significant benefits in fine-tuning. This improvement can be attributed to the critical role these weights play

| Target Modules | Accuracy |
|---|---|
| Q | 64.19 |
| K | 64.31 |
| V | 63.71 |
| Down | 62.63 |
| Up | 62.91 |
| Q,K,V,Down,Up | 63.10 |

Table 5: Accuracy comparison of placing LoRA at different target modules.

in attention mechanisms, highlighting the importance of carefully selecting target modules for optimization.

## 5 RELATED WORKS

**Generative foundation models.** Generative deep learning models pre-trained on large datasets are called generative foundation models (Bommasani et al., 2021). These foundation models can be applied to downstream tasks by fine-tuning. Advanced generative foundation models in natural language processing (NLP) such as GPT (Brown et al., 2020; Ouyang et al., 2022) and LLaMA (Touvron et al., 2023a) model have shown great success in assisting and generating human-like text across a wide range of topics. These generative language models can also be applied to many practical downstream tasks, such as education (Kasneci et al., 2023) and healthcare (Thirunavukarasu et al., 2023). Another kind of generative foundation model that has developed maturely is the diffusion model (Ho et al., 2020; Rombach et al., 2022). The diffusion model works well in various computer vision tasks such as text-to-image generation (Everaert et al., 2023) and image editing (Kawar et al., 2023).

**Efficient fine-tuning methods.** Efficient fine-tuning methods aim to reduce the number of trainable parameters to save the GPU memory and training time during fine-tuning large-scale models. Some PEFT methods freeze most of the parameters in the model and only fine-tune specific modules, e.g., BitFit (Zaken et al., 2021) fine-tunes only the bias of the model, which significantly saves the GPU memory. However, it cannot be executed on models without bias parameters. (Houlsby et al., 2019) and (Pfeiffer et al., 2020) add adapter layers between transformer blocks. These methods accelerate fine-tuning by transferring knowledge from adapter layers pre-trained on general tasks. LoRA (Hu et al., 2021; Liu et al., 2024b) is the most popular adapter for fine-tuning large foundation models. It adopts the product of two small matrices to represent the full gradient during fine-tuning. It can reduce the number of trainable parameters by 10,000 times and the GPU memory requirement by 3 times. Some adaptive algorithms work together with LoRA that can dynamically adjust the number of trainable parameters to fit specific needs. For instance, AdaLoRA (Zhang et al., 2023) adaptively allocates the trainable parameters to fit the GPU memory budget. These adaptive algorithms on LoRA require heterogeneous LoRA configuration when implemented in federated fine-tuning.

## 6 FUTURE WORK

Our proposed SHARELORA method is orthogonal to other LoRA variants, suggesting that future work could explore combining SHARELORA with existing PEFT methods. For example, LoRA+ (Hayou et al., 2024) sets different learning rates for the adapter matrices $A$ and $B$. By adjusting the learning rate ratio between these two matrices, the efficiency and performance of fine-tuning can be significantly improved. Specifically, LoRA+ sets the learning rate of $B$ to be $\lambda$ times that of $A$. This technique could easily be integrated with SHARELORA. Additionally, DoRA (Liu et al., 2024a) is a novel PEFT method that enhances LoRA by decomposing pre-trained weights into magnitude and direction components for more precise fine-tuning. This weight decomposition allows DoRA to optimize the magnitude and direction of weights separately. SHARELORA could also be combined with DoRA to share the magnitude and direction components among redundant layers. In summary, SHARELORA not only provides a robust solution for fine-tuning but also offers the flexibility to integrate with advanced PEFT methods, opening new avenues for research and potentially leading to significant gains in PEFT performance and efficiency.

## 7 CONCLUSION

In this paper, we present SHARELORA, a parameter-efficient fine-tuning method that determines which layers share the LoRA weights based on layer redundancy. SHARELORA leverages the cosine similarity between each layer's representations to ascertain redundancy. Utilizing a greedy algorithm, we maximize the sharing of LoRA weights while adhering to a predefined similarity threshold. This approach effectively reduces the number of trainable parameters. We conduct extensive experiments on large language models and multi-modal vision-language foundation models. The results demonstrate that SHARELORA achieves comparable or superior performance to existing PEFT methods, while using only 80% of the trainable parameter budget.

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
