# OpenReview forum: "ShareLoRA: Less Tuning, More Performance for LoRA Fine-tuning of LLMs"
_ICLR.cc/2025/Conference — ICLR 2025 Conference Withdrawn Submission_

### Official Review · Reviewer_Bk3n · 2024-10-17

**Soundness:** 2
**Presentation:** 2
**Contribution:** 2
**Rating:** 3
**Confidence:** 4

**Summary:**

The proposed method groups the layers of an LLM such that layers in each group have pairwise similarities above some threshold. Then the same LoRA adapter is learned for all layers in each group.

**Strengths:**

There is some saving in the number of LoRA parameters due to sharing the LoRA adapters.

**Weaknesses:**

It is the opinion of this reviewer that even if we take the claimed contributions of the paper at face value, the paper is still below the acceptance threshold.  (1) A 20% reduction in parameters is not very impressive, considering the previous work mentioned by the authors on layer duplication, parameter sharing, etc. (2) The proposed method is a greedy heuristic and not optimal.

A minor remark on point (1) above, the word "surprisingly" in the 3rd paragraph of page 2 is a bit subjective. Given the existing literature, a 20% saving in parameters without loss of accuracy and even some gain is almost expected. On the same page, "improves finetuning efficiency" -- do you mean the reduction of the parameters and memory or also runtime? I did not find any evidence for this.

Eq.2 simply counts the number of layers that have shared LoRA. However, if we have a set of L/2 pairs then the saving is much lower than in the case of one large set of L layers. The criterion can be such that it is tied more closely to the goal of reducing the number of parameters.

Eq.3 is sloppy and should be defined more carefully "for each set s in S" and without the typo of x_i twice.  I expect it to be "s.t. ∀s ∈ S, ∀i, j ∈ s, c(xi, xj) ≥ ϵ" (suggestion by the ICLR review feedback robot)

Eq. 4 defines cosine similarity but does not tell us anything about what is actually implemented. According to the subsequent text the representation of the last token is used. I guess that the same token representation is used also in Figure 1 (two pages before Eq. 4), but this is not mentioned at all.

Algorithm 1 is also sloppy. i not in S is not defined, for example. I guess the authors mean that 'There is no set in S that contains i'. (suggested by the ICLR review feedback robot) or better yet as an equation "i not in the union of s in S"

See also questions 1--3 below for further weaknesses.

**Questions:**

1, Why have different thresholds at different experiments?
2. Are the added ranks according to the ratio at the bottom of page 5 used in any of the experiments? what are the results?
3. Why not distribute the added ranks in that ratio between the groups S_i according to the size of the group?
4. Why not have a criterion that directly maximizes the parameter reduction, instead of Eq. 2?
5. Out of all possible greedy heuristics, why was the one in Alg. 1 selected?

---

### Official Review · Reviewer_rfLQ · 2024-11-03

**Soundness:** 3
**Presentation:** 4
**Contribution:** 3
**Rating:** 5
**Confidence:** 3

**Summary:**

This paper introduces a simple yet elegant method for parameter-sharing LoRA method by computing the similarity matrix of transformer layers and grouping the layers together based on the similarity threshold. Then, similar layers are grouped to share lora adapters. Overall, the paper is well-written and easy to follow. The method reduces the number of trained parameters while the performance is still competitive. The similarity computation lacks novelty because this similarity computation method is widely applied in layer pruning methods. Regarding this method, I have some questions and concerns; see details below. Also, the paper does not describe some technical details.

**Strengths:**

1. The paper is well-written and easy to follow
2. The method is simple and elegant. This method can potentially apply to a wide range of cases
3. The experiments and ablations are promising. This method reduces the number of parameters while the performance is competitive.

**Weaknesses:**

1. One of my concerns is that this approach's number of training parameters is dynamic and needs to be estimated case by case. Therefore, for parameter-efficient fine-tuning, the resource allocation needs to be dynamic for large-scale fine-tuning structures.
2. This paper only evaluates on the LLaMA 7b or 8b models. The number of training parameters of the original Lora method is small, and reducing 20% of the parameters may not change the minimum GPU requirements. This approach might benefit multiple LoRA serving but this paper does not provide comprehensive analysis for the memory during training or serving.

**Questions:**

1. Does this approach follow a similar pattern when applying this approach to larger models such as 70B or 405B models?
2. In this implementation, do Q, K, and V share the same Lora across layers, or do they share a separate Lora?
3. Have you testing the instruction fine-tuning across a combined SFT set to see whether this method can be applied to the general instruction settings? Or this method is only tested on specific fine-tuning setting on each specific training set of the datasets?

---

### Official Review · Reviewer_gqAc · 2024-11-03

**Soundness:** 3
**Presentation:** 3
**Contribution:** 1
**Rating:** 3
**Confidence:** 4

**Summary:**

The paper introduces SHARELORA, a parameter-efficient fine-tuning (PEFT) approach designed to enhance LoRA’s efficiency by leveraging layer-wise redundancy in large language models (LLMs). SHARELORA identifies similar representations across layers and enables the sharing of LoRA modules, reducing the number of trainable parameters while maintaining or enhancing performance.  SHARELORA achieves up to 23% fewer parameters with comparable or improved performance. It demonstrates versatility by effectively fine-tuning both language and vision-language models.

**Strengths:**

1. SHARELORA introduces an approach to fine-tuning by enabling parameter sharing across redundant layers in LLMs, which reduces parameters without additional computational cost.

2. The paper is well-supported with experiments across language and multimodal models. The authors compare against strong baselines, showing that SHARELORA can match or even improve performance while using fewer trainable parameters.

**Weaknesses:**

1. SHARELORA focuses solely on inter-layer redundancy without addressing potential intra-layer redundancy, where individual layer components (e.g., attention heads or submodules) might also share information. Ignoring intra-layer redundancy can lead to missed opportunities for additional parameter reduction, especially in high-dimensional models where significant duplication exists within single layers.

2. The method lacks a detailed analysis of where redundancy is most prominent across the model. For example, quantifying layer similarity through metrics like entropy or similarity scores could better guide decisions on which layers should share parameters.

3. The paper would benefit from more granular ablation studies, such as testing different similarity thresholds, rank configurations, and sharing strategies to assess their impact on performance.

**Questions:**

1. Could the authors clarify how they selected the similarity thresholds for determining layer redundancy?

2. Given that redundancy can occur within layers, did the authors consider intra-layer sharing (e.g., across attention heads or feed-forward components within a layer)?

3. Have the authors considered implementing shape adaptation mechanisms, such as transformation techniques that allow shared parameters to fit across layers of varying dimensions?

4. Have the authors considered including these analyses to substantiate the choice of layers for parameter sharing?

5. Could the authors provide more detailed ablation studies on the effects of varying ranks, similarity thresholds, or different parameter-sharing strategies?

---

### Official Review · Reviewer_Z2VG · 2024-11-04

**Soundness:** 2
**Presentation:** 3
**Contribution:** 2
**Rating:** 5
**Confidence:** 4

**Summary:**

In this paper, the authors proposed ShareLoRA, a new PEFT method that allows the LoRA modules to be shared among multiple layers. A greedy algorithm was used to find these layers with similar representations. Experiments on multiple datasets show some good performance in certain tasks over several baseline methods.

**Strengths:**

1. Multiple experiments were conducted on several tasks with comparison to a few baseline methods.
2. Writing in general is good and easy to follow.

**Weaknesses:**

1. There is already very limited tunable parameters with LoRA, I am not sure further reducing it by ~20% is worthwhile given the complexity introduced in the proposed approach and its not convincing performance (i.e., the result from Table 2 and 3).
2. The need of sample dataset D∗ makes the proposed algorithm less applicable unless one common calibration dataset can be used for all the different tasks which I did not see from the current draft.
3. The performance seems very sensitive to the hyper-parameter similarity threshold from Table 4. Are there any principles or rule of thumb to set this parameters for different models/tasks?
4. Typo. "LoRA LoRA" in line 061 should be "LoRA".

**Questions:**

Overall, I am not quite convinced that the proposed approach is widely applicable. Please refer to the weakness section for the rationals and provide rebuttal accordingly.

---

### Official Review · Reviewer_Jrck · 2024-11-04

**Soundness:** 3
**Presentation:** 2
**Contribution:** 3
**Rating:** 5
**Confidence:** 4

**Summary:**

In this paper, the authors aim to leverage the redundancy present in pre-trained foundation models to further reduce memory requirements in LoRA. The authors propose the ShareLoRA to achieve this goal. The proposed ShareLoRA comprises two key components: (1) layer similarity computation; (2) LoRA sharing.

**Strengths:**

(1) The direction of reducing memory requirements in LoRA by layer sharing is interesting.

(2) The proposed method is sound.

(3) The paper is easy to follow.

**Weaknesses:**

(1) The authors should check their writing. In Line 061, During fine-tuning, LoRA LoRA. In Line 182, the name of the proposed method is ShareLoRA, not AutoLoRA.

(2) The authors should add experiments on Llama-13B to demonstrate the scalability of the ShareLoRA.

(3) the authors should add experiments on NLG tasks to demonstrate the generalizability of the ShareLoRA.

**Questions:**

Please see weaknesses.

---

> ### Comment · Reviewer_Jrck · 2024-11-27
>
> As the authors do not make responses, and I checked other reviewers' comments, I decide to lower my rating.

---

### Note · Authors · 2024-12-09

I have read and agree with the venue's withdrawal policy on behalf of myself and my co-authors.